# Directions for Anarchist Studies †

Kathy E. Ferguson

Departments of Political Science and Women's, Gender, and Sexuality Studies, University of Hawai'i at Mānoa, Honolulu, HI 96822, USA; kferguso@hawaii.edu
† An earlier version of this paper was presented on a roundtable at the British Political Science Association annual conference in 2022.

**Abstract:** Anarchism is a fertile site for nurturing the sorts of encounters that feminists have called intersectionality. Anarchism and intersectionality share the goal of critically examining familiar as well as emergent flows of power and meaning, and understanding their relations to one another. This paper focuses on three compelling directions for anarchist studies: Indigenous anarchism, anarchism developing with new materialism, and anarchism emergent in radical book arts. Each thread has established roots while also moving in new directions. Anarchist encounters with Indigeneity, new materialism, and book arts resonate with each other: they can foster "a commitment to the particular" through which we can immerse ourselves in rich and dense worlds where specific Indigenous theories and practices, detailed encounters with non-human things, and particular artistic + intellectual productions of materials can emerge.

**Keywords:** anarchism; Indigeneity; new materialism; book arts

## 1. Introduction

I see anarchism as a space of invitation. Anarchist philosopher David Wieck characterizes anarchism as "the *generic* social and political idea that expresses negation of all power, sovereignty, domination, and hierarchical division, and a will to their dissolution" [1] (p. 6, italics in original). Literary scholar Saidiya Hartman succinctly remarks, "Anarchism is an open and incomplete word, and in this resides its potential" [2] (pp. xv–xvi). Putting these ideas together—anarchism as opposition to hierarchies, and as perpetually incomplete—suggests that anarchism is a fertile site for nurturing the sorts of encounters that feminists have called intersectionality. Critical race theorist Kimberlé Crenshaw characterizes intersectionality as an open-ended intellectual tool for simultaneously apprehending "both the structural and the dynamic aspects" of power relations [3] (p. 230). Reflecting on intersectionality's travels since she coined the term in 1984, Crenshaw highlights "the interpretive, creative, highly contested and sometimes perilous work of integrating insurgent knowledge into established, often conservative discursive communities" [3] (p. 224). Anarchism and intersectionality share the goal of critically examining familiar as well as emergent flows of power and meaning, understanding their relations to one another. The space of invitation in anarchism beckons critical political thinkers and activists to weave new threads into the mix, cultivating fresh relations.

Anarchism's openness to intersectional thinking is not new. Emma Goldman, to take one example, was an early analyst of the mutual interactions among gender, sexuality, class, and the state in capitalist society [4] (pp. 16, 250, 261). Still, intersectional anarchism seems to be "taking off" now and producing a rich cacophony of elaborations: Indigenous anarchism, anarcho-blackness, anarcha-feminism, queer and trans anarchism, ecological anarchism, anarchic species relations, and no doubt others as well. Here, I will focus on three directions of analysis and action that I hope continue to develop within the parameters of anarchist studies: Indigenous anarchism, anarchism developing with new materialism, and anarchism emergent in radical book arts. The first two already have rich intellectual

and political grounding upon which to build, while the third is less familiar but nonetheless has both historical precedent and contemporary expression.

Each of these directions of anarchist study offers rich opportunities for historical depth and empirical detail. In their reflections on anarchist studies in 2009, philosopher Ruth Kinna and political theorist Alex Prichard rightly warn against "an endless celebration of a few de-historicized and de-contextualized principles" [5] (p. 271). Environmental Studies scholar Adam Lewis similarly laments "a dangerous flattening of social conditions" that can come when anarchists eschew all hierarchies equally without "grasp[ing] the nuances, complexities, and particularities of their sites of struggle" [6] (p. 149). Instead of general exhortations about the compatibility or incompatibility of anarchist ideas with other philosophies of liberation, I hope for work that builds on deep dives into relevant historical and empirical evidence. A friend calls this "ass in the grass" research: thorough explorations of texts, artifacts, and archives; patient, attentive interviews and oral histories; extensive observation and ethnographic participation in movements or communities. We already have good models: Paul Avrich set the gold standard for broad and careful historical research in *The Modern School Movement* [7] and a remarkable set of interviews spanning several decades in *Anarchist Voices* [8]. Histories from below such as Chris Ealham's *Anarchism and the City* [9] and Kenyon Zimmer's *Immigrants against the State* [10] build detailed stories by digging deeply into archives. Meticulous participant observation in contemporary movements, such as Sarah Fessenden's fine dissertation on Food Not Bombs [11], eschews vague generalities in favor of a thorough, careful analysis of what they do, how, and why they do it.

Anarchist encounters with Indigeneity, new materialism, and radical book arts resonate with each other. They can foster something like "a commitment to the particular" through which we can immerse ourselves in rich and dense worlds where specific Indigenous theories and practices, detailed encounters with non-human things, and particular artistic + intellectual productions of materials can emerge. Each direction of thinking moves us a bit away from conventional western expectations that good philosophy must be abstract, universal, and systematic, and more toward practices grounded in immediate, specific, and narrative encounters. Each area offers a version of located theory, that is, detailed analysis grounded in particular circumstances: in the lands and waters of Indigenous worlds, the lively actancy of things, and the specific craft aesthetics of book artists.[1] In the remainder of this essay, I look briefly at each of these, suggesting ways in which each can enrich anarchist studies.

## 2. Indigeneity

Indigenous encounters with anarchism bring a multitude of fresh energies. In their magisterial counter-history *The Dawn of Everything*, David Graeber and David Wengrow center what they call "the indigenous American critique of European society" in order to dislodge the hold of colonial agendas on historical thinking [12] (p. 61). This is an essential move for any Euro-American-based theory, including anarchism, given the utter ubiquity of colonial assumptions about history as a long, grand evolutionary ladder leading from simple to complex societies, from "foragers liv[ing] in a state of infantile simplicity" to settled farmers and property owners [12] (p. 148). Graeber and Wengrow emphasize the dense hegemony of the evolutionist frame: they mildly note a "lack of alternative terminology" to imagine histories outside of the metaphors of growth and decay that keep us thinking that there is only one story to tell [12] (p. 449). This hegemony has constricted anarchism as well as the more mainstream Euro-American theories. For example, both Peter Kropotkin and Louise Michel struggled against the pervasive assumption that the Siberian peasants in Kropotkin's geographical studies and the Kanak peoples of New Caledonia with whom Michel worked during her incarceration there, following the crushing of the Paris Commune, were lesser figures on a grand historical scale; but both writers groped unsuccessfully for a language in which to "drop the teleological habit of thought" [12] (p. 382). Kropotkin's language in *Mutual Aid* reflects the tactics he marshalled to contest the progress

narrative: he talked about "so-called primitive" and "so-called civilized" people to question the categories; he used scare quotes around "barbarism" and "savages" to notify readers of their unsavory implications [13] (pp. 86, 89, 97, 110, 112, 115, 118, 132). Similarly, Michel was a staunch ally of the New Caledonian rebels in their struggle against the French, but she had difficulty moving away from the language of "uplift" bestowed on a child-like people [14] (p. 224). Both anarchists regularly reversed the prevailing terms, claiming true civilization for the older societies while Europe and Euro-America exemplified savagery. Yet simply reversing the terms does not change the frame. Historian Carolyn Eichner's trenchant observation about Michel's difficulty in getting outside the dominant frame rings true for Kropotkin as well: "…she could not fully separate herself from aspects of the supremacism of white European intellectual culture" [14] (p. 224). "Unpack[ing] [our] Euro-centric baggage" is not just a matter of intention or will: it requires the continuing work of taking the perspectives of Indigenous peoples onto the world and, as Adam Lewis insists, "centering them in our work and our movements" [6] (pp. 166, 167).

Now, I suggest, anarchist scholars have a better chance of contesting this frame because today there is an explosion of writing, speaking, cultural productions, and political organizing expressing the worldviews of Indigenous peoples. Indigenous voices have critiqued western society before, of course. Graeber and Wengrow's momentous account of the scathing review of French society generated by Indigenous observers, particularly by a man named Kandiaronk of the Wendat Confederation, documents the impact of this encounter on earlier European writers including Montesquieu, Diderot, and Voltaire [12] (pp. 49–58). Acknowledging this history, it is still the case that many European and Euro-American thinkers, as well as the larger publics they address, imagine Indigenous societies as properly sorted at the beginning of a grand, one-way evolutionary ladder. Yet, today's growing critical mass of Indigenous scholarship, art, music, and literature across many academic fields and political landscapes offers a propitious opportunity. Contemporary Indigenous scholars and activists are challenging hegemonic views by writing and speaking from the resurgent auspices of Indigenous studies, and anarchist studies has the opportunity to learn with them.[2] While such cooperative ventures are likely to be tumultuous, they are suggestive. Kanaka Maoli (Native Hawaiian) scholar Kahala Johnson suggests we "acknowledge the cacophony" in such encounters in order to explore them further [15] (p. 706). I briefly take up two examples here: challenges to dominant understandings of time and to hegemonic practices of governance.

Both anarchism and Indigeneity are attached to what literary scholar Mark Rifkin calls "discrepant temporalities" [16] (p. 3). From colonial perspectives, Indigenous life is seen as lost in the past, with its adherents either vanished or fixed in a vague premodern haze. Framing time in relation to land, water, other animals, and responsibilities to prior and future generations, Indigenous time struggles against colonial time, with its investments in progress and civilization. Anarchist time similarly struggles against state time; while state-centric time views anarchism as hopelessly utopian, anarchists instead find practices of mutual aid and prefiguration already seeded into our pasts and our presents. British anarchist Colin Ward sees the elements of anarchism as already visible "in the interstices of the dominant power structure. If you want to build a free society, the parts are all at hand" [17] (p. 20). Indigeneity and anarchism intersect in their untimeliness, refusing to be relegated either to an allegedly lost past or unachievable future.[3]

Anarchism and Indigeneity imagine self-governance differently but in ways that can resonate with one another. The difference appears obvious: after all, David Wieck named "sovereignty" as one of the power arrangements anarchists must unitarily oppose, while Indigenous peoples may characterize their struggle for self-government as a sovereignty movement. Kanaka Maoli scholar J. Kēhaulani Kauanui analyzes the conflict: "Anarchists (who are non-Indigenous) may understand 'sovereignty' as always already a form of domination through a state monopoly on the use of violence against its citizens" [18] (p. 17). Yet there is more than one way to understand sovereignty, just as there is more than one way to imagine time, and Indigenous sovereignty activists "are often referring to their collective

inherent authority to govern and assert their self-determination as politics" [18] (p. 18). Commenting on the related concept of nationhood, Kauanui points out that "for Indigenous peoples the assertion of nationhood is about survival *as peoples,* given the endurance of colonial domination" [18] (p. 17). Indigenous struggles for sovereignty and nationhood challenge anarchists to envision those not necessarily as a "bid for state power" but as a source of practices of self-governing [18] (p. 17). In a similar vein, Kwakwaka'wakw artist Gord Hill notes that Indigenous visions may feature extended families and hereditary leadership, while anarchists focus more on personal freedom and intentional communities, yet he too sees "Indigenous sovereignty [as] a path towards liberation from the state and capital, not just for Indigenous peoples but for non-Indigenous people as well" [19] (p. 111).

Kanaka Maoli scholar Noelani Goodyear-Kaʻōpua offers stories from the Hawaiian sovereignty movement in which resistance was "constituted through direct action for aloha ʻāina [love of the land] and collective decision-making" [20] (p. 138). Goodyear-Kaʻōpua suggests that, rather than following the directives of leaders or seeking the permission of authorities, Hawaiian sovereignty activists create change by living differently: "It is in the process of these mobilizations, rather than in the final positions enunciated, that revolutionary potential is located" [20] (p. 133). Johnson continues this inquiry by exploring indigenous demands for the deoccupation and reestablishment of pre-colonized nations, in conversation with anarchist critiques of all states. Conversations between anarchist studies and Indigenous studies have a lot to offer.

### 3. New Materialism

Inquiry into relations between humans and non-human others can also open anarchism to new directions. New materialism is a theoretical cousin to network theory and assemblage theory, which has been recruited by numerous writers on anarchism, including political scientist and historian Benedict Anderson's analysis of global anarchism as a "vast rhizomal network" [21] (p. 160) and historian Andrew Hoyt's use of network analysis to study the anarchist journal *Cronaca Sovversiva* [22]. New materialists push network theory and assemblage theory in the direction of "thing power" [23] (p. xxii). Political theorist Jane Bennett explains the liveliness of matter as "the capacity of things—edibles, commodities, storms, metals—not only to impede or block the will and designs of humans but also to act as quasi agents or forces with trajectories, propensities, or tendencies of their own" [23] (p. 72). Summarizing new materialism's general direction, philosopher Arianne Conty writes,

> Instead of separating the active human subject from a world of passive objects, the world of human culture from a background of brute nature and the human being from all non-human life-forms, new materialism is the name that has been given to the scholarship devoted to replacing modern dichotomies with a processual and contextual understanding of the human being as part of a unified theory of life. Though the scholars grouped under this heading are extremely diverse, they all seek to celebrate the materiality of our world, a materiality understood no longer as inert and passive, but rather as vibrant and full of agency. [24] (p. 103)

Non-organic matter is brought into the intersectional mix as lively, as having actancy, and as being capable of acting and being acted upon.

New materialism is heavily indebted to Lucretius, Spinoza, Gilbert Simondon, and other earlier process-oriented materialisms. (Unsurprisingly, this suggests that new materialism is not so new but has older manifestations in European thought.) Political theorist James Martel and philosopher Chiara Bottici have both drawn usefully on Spinoza to undo the hegemony of possessive individualism and what Martel calls "archism", "a form of politics based on rule and hierarchy", in favor of anarchism [25] (p. 1). Bottici works with Simondon's theory of individuation as the best way to understand how human subjects come to be outside the determinism of prevailing hierarchies [26]. Catherine Malabou's work interweaves philosophy and biology to expand anarchist ideas of solidarity and mutual aid outside the frame of "debt, obligations, duty, or guilt" [27] (p. 2).

I have found new materialism indispensable in analyzing anarchist print culture, bringing presses, ink, paper, and letters into the picture as participants in the making of anarchism.[4] Anarchism's remarkable history of writing and publishing takes on a new coherence when the materiality of the presses' "reek and clamour", the printers' laboring bodies, and the pages' physical architecture come into view as helping to *produce*, not just reflect, anarchism [28] (p. 15). Anarchist publications were not simply reporting on politics that happened elsewhere; rather, the writing, printing, distributing, and archiving of publications were a generative happening of anarchism. Similarly, scholars of Malibou's work have engaged classical anarchist ideas, notably Kropotkin's ideas of mutual aid, in light of Malibou's suggestions for political resistance in nature/culture. Malibou rethinks the opposition of biology with discursivity that has propelled much poststructural anarchism, instead offering "the profile of a reelaboration of anarchism today" based on her reading of mutual aid through the lens of the plasticity of our brains and genomes [29] (p. 15). She builds on her historicizing of biology to offer "unchained solidarity" as a new materialist-inflected anarchist practice [29] (p. 19).

New materialism can forge rich connections with the more familiar materialism of Marxist-inspired class/state analyses, as well as with Indigenous understandings of living forces in non-human entities and with the merger of art and craft in radical book arts. Anarchists have typically been hands-on: they have created alternative institutions, including schools, cafes, publications, bookstores, workshops, theaters, libraries, unions, and independent communities. All these practical projects have entailed working with non-human matter, and in turn being worked on by that matter, to make things happen. Anarchist studies could further investigate these sites of creative practice. Bringing non-organic things into intersectional relations with other flows of power encourages us to theorize them not as a static background but as vibrant partakers in political relations.

## 4. Radical Book Arts

During the late 19th to mid-20th centuries, William Morris and Walter Crane in England and Joseph Ishill in the U.S. led a movement to rescue book arts from industrial degradation and bring fine printing into radical politics. They embodied the anarchist commitment to combining intellectual and manual work, beauty and function, art and craft. In one of hundreds of letters sent to Ishill by appreciative readers, Forest Frazier of Huntsville, Missouri, wrote to thank Ishill for his work: "Type, illustrations, paper & binding fit the text perfectly.... A beautifully printed page gives the same thrill as a beautiful painting or sculpture. Fine printers, like you, have lifted a craft into the realm of great creative art" [30] (p. 284). Readers of publications printed by Morris and Ishill were simultaneously offered intellectual, political, and aesthetic rewards. The content of the material interacted with the look and feel of the pages to create both intellectually powerful and aesthetically moving encounters. Readers both learned about the world and were invited to encounter that world in fresh ways.

PM Press's striking new edition of Kropotkin's *Mutual Aid* revisits the tradition that art historian Allan Antliff calls "the anarchist politics of the book" [31] (p. 254). *Mutual Aid* is a good choice for extensive illustration in that Kropotkin too wanted art to be "bound up with industry by a thousand intermediate degrees..." [31] (p. 255). Contemporary illustrator N.O. Bonzo is inspired by the Arts and Crafts movement initiated by Morris and Crane, which aimed to unite art, artisanship, and production so that ordinary objects combine utility with beauty. Bonzo lays out the page in much the same way as did Morris, with ornate borders of interlocking leaves and flowers, often with dramatic black backgrounds. Side panels offer graceful scenes of working people laboring and relaxing, building a new and just world together. Antliff observes that such illustrations heighten the visual presentation of anarchist values through "the rhythmic interplay of arabesque lines and compositional arrangement" [31] (p. 262). Antliff clarifies the political power of beauty in the illustrative work of Morris, Crane, and Bonzo:

Stimulating the imagination, beauty was an agitational, desire-producing social force…the aesthetic attractiveness of a book could simultaneously amplify its critical content by virtue of its beauty, and the seductive power of that beauty. It could, in effect, prefigure a world transformed, even as it points to 'evils of all kinds' and the need for transformation. [31] (p. 258)

Another radically illustrated anarchist book is Charlie Allison's analysis of the Ukrainian anarchist leader Nestor Makhno, *No Harmless Power* [32]. The cover by Bonzo and Kevin Matthews is similar to Bonzo's Morris-inspired work on *Mutual Aid*: a dramatic black background decorated by snaking vines and flowers around the central figure of Makhno, kneeling in a field and caressing a flower. A few other illustrations follow this aesthetic form, while still others reproduce old photographs. Most striking of all are the fifteen portraits of "Anarchists You Should Know" at the end of the book. In startling black and white, these full-page figures emulate the fine lines and severe contrasts of old-fashioned wood engraving, a favorite method of illustrating anarchist publications in earlier times. Blinding white faces take shape against dramatic black shadows, often merging without borders into glossy black backgrounds. This vivid collection of radical portraits produces a fitting aesthetic for these little-known figures. It is as though each face looms suddenly out of the darkness, after all this time, to meet us.

A third recent contribution to radical book arts is Charles Overbeck's lovely production *The Tramp Printers: Forgotten Trails of Traveling Typographers* [28]. Overbeck's printers travel across the dramatic black cover and nestle on the stippled pages. He constructs the page much as Morris and Ishill did: his generous margins let the text breathe, while illustrations, illuminated letters, and graceful fonts encourage readers to pause and notice the page as a setting for staging the text in space as well as presenting the words themselves. These books provide more than pictures to illustrate ideas, although they do that as well. Bonzo sees book arts, along with street arts, as expressing anarchism's lifeworld, "accessible and immediate, an active and participatory part of everyday life" [33] (p. 270). Rich illustrations, generous layouts, diverse fonts, illuminated letters, intriguing fleurons (small figures marking the beginnings and endings of texts), printers' colophons, and other elements of the page's architecture do more than structure texts: they act on readers, inviting them to pause, skip ahead, leaf back, and reflect. I have used Bonzo's new illustrated edition of *Mutual Aid* in class, with gratifying results: students happily sat with the book, dwelling in it, calling each other's attention to the architecture of each page. They lingered over the vivid illustrations and traced the patterns in the wide borders. Many of them found the task of reading more pleasant, the pages more inviting. Like Morris, Crane, and Ishill, contemporary book artists create the form of the book in conversation with its contents. They reach people with more than ideas, and more than images, but with an interweaving of both together.

## 5. Conclusions

Indigeneity, new materialism, and radical book arts resonate usefully with one another: Indigenous theory, as articulated here by Kanaka Maoli writers, is already strongly materialist, in the sense that it is place-based. As Maya L. Kawailanaokeawaiki Saffery writes, "Kanaka understand that we come from the land itself… We believe that the land, the sea, the sky, and all creatures that exist in the universe are all our kupuna as much as our human grandparents" [34] (p. 109). Goodyear-Ka'ōpua's concept of "land-centered literacies…that include facility in multiple languages, human and nonhuman" [35] (p. 35) resonates with Bennett's appeal that we "afford voice to vibrant materials whose first language is not words" [36] (p. xxiv). The two materialisms draw on different ontologies: Kanaka Maoli thought discerns spirit in matter and familial relations in non-human nature, while new materialist ontologies do not usually locate spirit within matter, but instead find matter itself to be lively. Yet, the two materialisms can recognize each other and usefully work together. Further, Adam Barker and Jenny Pickerill point out that indigenous geographies can be an opportunity for non-indigenous anarchists to "find their own new way of looking

at—and being in—place" [37] (p. 1715). Similarly, anarchist book arts embody a vigorous materialism, in that writers, artists, and printers create intellectual and aesthetic objects that in turn create the people who make and read them.

These three directions of research are intersectional in the most generous sense of intertwining diverse threads that take their emergent meanings from their dynamic relationships to one another. Eschewing hierarchies of thought in favor of networks of interactions, these directions of inquiry can expand anarchism's space of invitation. Critical conversations among these theories and practices can allow frictions to be articulated while still learning from one another.

**Funding:** This research received no external funding.

**Acknowledgments:** My thanks to my fellow panelists and attendees at the 2022 panel on the future of Anarchist Studies at the British Political Studies Association annual meeting, and to three insightful anonymous reviewers.

**Conflicts of Interest:** The author declares no conflict of interest.

## Notes

1. For further discussion of located theory as situated, event-based, and engaged, see [4], pp. 5–6, 67–68, 220–221.
2. In this relatively brief discussion of anarchism and indigeneity, I want to recognize patterns without overgeneralizing about Indigenous societies. I rely most heavily on Kanaka Maoli (Native Hawaiian) scholars and activists because I work and live most closely with them.
3. For more extensive discussion of untimeliness in anarchism and Indigeneity, see [14].
4. For more extensive discussion of new materialism as a method for analyzing anarchist print culture, see K. Ferguson, *Letterpress Revolution: The Politics of Anarchist Print Culture* (Durham, NC: Duke University Press, USA, 2023).

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
