# Peer review of "Directions for Anarchist Studiesâ€"

_philosophies, doi:10.3390/philosophies8050088_

Round 1
Reviewer 1 Report
You may wish to also consult Mohawk scholar Taiaiake Alfred's Wasase: Indigeous Pathways of Action and Freedom (2005), which discusses the interface between Indigeniety and anarchy. Also, the historical grass roots interface of anarchist/Indigenous thought in Canada has been examined in a dialog between Allan Antliff and Indigenous anarchist sovereigntist Gord Hill in a recent issue of Anarchist Developments in Cultural Studies. This article discusses the Indigenous anarchist sovereigntist Mel Bazil's views at length, for example. If one is to address 'sitting in the grass' theory, than the voices of these activists might merit a paragraph.
Author Response
Thank you so much for your comments. We have finished the revision with the revised version attached.

Reviewer 2 Report
This article is tight well-crafted and makes a strong contribution to the field. The three directions the author suggests (Indigenous anarchism; anarchism developing with new materialism; and anarchism emergent in radical book arts) are fruitful and generative.
Minor issues that need attention:
“Critical legal theorist Kimberly Crenshaw” should read: “Critical race theorist Kimberlé Crenshaw” since she helped found the field of critical race theory as a corrective to critical legal studies, and please note the spelling of her first name.
P3 “ Indigenous people” should read: “Indigenous peoples” (i.e. plural) for people when referring to them as collective polities rather than any mere group of Indigenous individuals (the article has a couple spots where 'people' is correct but needs 's' on the end in at least one spot).
Also on page 3, the paragraph the begins with “Both anarchism and indigeneity are attached to what literary scholar Mark Rifkin..”needs a robust topic sentence rather than starting it off with a very lengthy quote. And it would be ideal to lay out a working definition of indigeneity.
Also on p3, “KÄ“haunani” should read “KÄ“haulani” and the lengthy article cited by her should be more fully engaged since it grapples with indigeneity as an analytic in a substantial way (and moves beyond anything specifically Kanaka Maoli). And in terms of Kanaka Maoli indigeneity, see:
Kauanui, J.K. 2019, “Anarchist Charges and the Politics of Hawaiian Indigeneity and Sovereignty,” HÅ«lili: Multidisciplinary Research on Hawaiian Well-Being, Volume 11, No.1, October 2019, pp207-213
The focus on Hawaiian indigeneity is strong, but the article could use a sentence or two to offer a rationale for that focus (as opposed to others, e.g. Native Americans, Ainu, Maori, Sami, et al).
Author Response

(The authors gave the same response as above.)

Reviewer 3 Report
This is an excellent article, which I very much enjoyed reading. The author has a command of the relevant scholarly literatures, writes well, and develops a clear and refreshingly original argument supported by compelling evidence.
As someone who works on all three of the areas examined, I can attest to the scholarly rigor and originality of the argument, which will make a very helpful contribution to anarchist studies and which should be published without question.
In more substantive terms, the article explores anarchist encounters with Indigeneity, new materialism, and radical book arts, and suggests that these three directions of research are promisingly intersectional in the sense that critical conversations cutting across such diverse fields of inquiry and practice can open up new horizons in anarchist studies. While none of the material discussed in each of the three sections of the paper is itself original, the juxtaposition of the three is fascinating, and indeed suggestive of promising new directions in anarchist studies. Notably, the author correctly emphasizes the 'commitment to the particular' common to all three: from the place-based animism of Indigenous cultures to the liveliness of matter emphasized in new materialism to the craft aesthetics of a William Morris or Charles Overbeck book.
In terms of constructive suggestions for improvement prior to final publication, it would be interesting and 'resonant' (to borrow a term from the paper) to see the author incorporate relevant insights from the now well-established but constantly evolving field of utopian studies, especially that fluid, vibrant, and intersectional aspect of the field focused on anarchist and anarcho-feminist indigenous ecotopianisms. See, for example, the chapter on 'Anarchism' in *The Palgrave Handbook of Utopian and Dystopian Literatures*, eds. Peter Marks, Jennifer A. Wagner-Lawlor, and Fatima Vieira, Palgrave Macmillan, 2022, 333-348; Jacqueline Lasky, 'Indigenism, Anarchism, Feminism: An Emerging Framework for Exploring Post-Imperial Futures', *Affinities: A Journal of Radical Theory, Culture, and Action* 5.1, 2011, eds. Glen Coulthard, Jacqueline Lasky, Adam Lewis, and Vanessa Watts, 3-36; Bill Ashcroft, *Utopianism in Postcolonial Literatures*, Routledge, 2017; and Leanne Betasamosake Simpson, *As We Have Always Done: Indigenous Freedom Through Radical Resistance*, University of Minnesota Press, 2017.
It might also be useful for the author to reflect further on why it is that 'a commitment to the particular' has been so marginalized in the history of Western political thought, including (even if perhaps to a lesser extent) in the anarchist tradition. There is some discussion of this point on page 2 of the article, but further consideration of this point would no doubt enrich the argument.
In sum, this is an excellent and unusually thoughtful and thought-provoking piece of work which I unhesitatingly and warmly recommend for publication.
Author Response

(The authors gave the same response as above.)
